# A Novel Quantification Method for Gene-Edited Animal Detection Based on ddPCR

**DOI:** 10.3390/biology14020203

**Published:** 2025-02-14

**Authors:** Kaili Wang, Yi Ji, Cheng Peng, Xiaofu Wang, Lei Yang, Hangzhen Lan, Junfeng Xu, Xiaoyun Chen

**Affiliations:** 1School of Food Science and Engineering, Ningbo University, Ningbo 215211, China; m15256105973@163.com; 2State Key Laboratory for Managing Biotic and Chemical Threats to the Quality and Safety of Agro-Products, Key Laboratory of Traceability for Agricultural Genetically Modified Organisms, Ministry of Agriculture and Rural Affairs, Hangzhou 310021, China; jymemory12138@163.com (Y.J.); pc_phm@163.com (C.P.); yywxf1981@163.com (X.W.); yanglei9203@163.com (L.Y.); 3Zhejiang Key Laboratory of Crop Germplasm Innovation and Utilization, Zhejiang Academy of Agricultural Sciences, Hangzhou 310021, China

**Keywords:** gene editing, MSTN, nucleic acid detection, ddPCR

## Abstract

The marketization of gene-editing products has attracted extensive attention from various countries as a means of guaranteeing the safety supervision of gene-editing products. In this study, a digital PCR detection method was established by analyzing MSTN gene-edited cattle with the aim of optimizing the reaction conditions of the method and performing tests on the specificity and sensitivity of the method. The method was then used to detect real MSTN gene-edited cattle samples. The outcomes of this study demonstrated that the method exhibited both adequate specificity and sensitivity, thereby enabling precise detection of gene-edited samples. The assay developed in this study can be utilized for the detection of this particular gene-edited cattle, thereby providing a technical foundation for the establishment of detection methods for gene-edited products.

## 1. Introduction

Gene editing is a technological tool that enables the alteration of an organism’s inherited traits by modifying its genome. The principal technologies are zinc-finger nucleases (ZFNs), transcription activator-like effector nucleases (TALENs), and regularly interspaced short palindromic repeats (CRISPR). Myostatin (MSTN) is a negative regulator of muscle growth [1]. Inactivation of this gene in animals may result in the phenomenon of “double muscle,” but it has no effect on the survival of animals. Deficiency or inactivation in animals may lead to an increase in the number of muscle cells, increased diameter of muscle fibers, increase in the number of muscle fibers, and overdevelopment of muscle. The successful development of gene-edited animals, including pigs [2,3], sheep [4], rabbits [5], and cattle [6,7], has been documented. The advantages of high meat yield, strong disease resistance, and weak aggressiveness inherent to gene-edited animals have led the United States to permit the marketing of gene-edited pigs and cows as edible meat. Furthermore, the safety of gene-edited products has become a topic of international concern, with the necessity for rigorous supervision being widely acknowledged. The regulatory stance towards gene-edited products differs across jurisdictions. In the United States and Japan, mandatory labeling of gene-edited products is not mandated, whereas in the European Union, labeling is not required for food products containing no more than 0.9% edited ingredients [8]. In China, however, gene-edited animal food products are required to display labeling indicating the presence of gene-edited animal tissues.

Current methods for detecting gene-edited products include next-generation sequencing (NGS) [9,10], PCR [11,12,13], and T7 endonuclease I [14,15]. However, these are generally semi-quantitative and less effective for processed foods with low DNA content or for precise quantification of gene editing [16]. Given the expected widespread use of gene-edited animals, establishing a quantitative detection method is essential. Droplet digital PCR partitions individual amplifications into separate compartments and detects end-point products, offering ultrasensitive and absolute nucleic acid quantification without a standard curve [17]. ddPCR has been extensively used for detecting genetically modified food [18,19] and shows higher accuracy [20] than other PCR-based techniques, making it suitable for detecting gene-edited products [21,22].

In this study, we selected MSTN gene-edited cattle with a code-shift mutation by knocking out 11 bp in the third exon of the bovine MSTN gene using CRISPR/Cas technology. We then proceeded to design three sets of primer probes, carry out the screening of specific primer probes, and optimize the method. The reaction conditions were established, and a method based on ddPCR was developed for the detection of gene-edited cattle. The results demonstrated that the method was specific and sensitive, with a lowest limit of detection (LOD) of 5 copies/µL, which enabled accurate detection of actual gene-edited samples. This method provides technical support for the detection of gene-edited products in the future.

## 2. Materials and Methods

### 2.1. Materials

The blood of MSTN-edited cattle and wild-type cattle, with universal primer sequences of forward CCTTGAGGTAGGAGAGTGT and reverse TCACATTCTCCAGAGCAGTA, were provided by Inner Mongolia University. The gene-edited cattle had an 11 bp deletion in the third exon of MSTN. Genomic DNA from six non-edited animals (cattle, sheep, pigs, chickens, quails, and pigeons) was stored in our lab.

### 2.2. Construction of Recombinant Plasmid for MSTN Gene-Edited Cattle

Genomic DNA from MSTN-edited cattle and wild-type cattle was extracted using a DNA extraction kit (TIANGEN, Beijing, China). The genomic DNA was amplified by PCR using a universal primer sequence, and the PCR products were sent for sequencing. The PCR product was sequenced and cloned into the pUC57 plasmid vector. Plasmid construction was completed by Shanghai Shenggong Biotechnology (Shanghai, China).

### 2.3. Plasmid DNA Extraction and Verification

Recombinant plasmid DNA was extracted using a plasmid DNA extraction kit and digested with SpeedyCut Hind III and EcoR I according to the manufacturer’s instructions. The digested product was purified using a gel recovery kit, and DNA integrity was assessed by 1% agarose gel electrophoresis. DNA absorbance was measured on a UV-visible spectrophotometer to evaluate plasmid purity.

### 2.4. Primer Design and Synthesis

Primers and probes for the target gene fragments were designed using Primer 5.0 based on sequencing results (Figure 1). All primers and probes were synthesized by Sangon Biotech (Shanghai, China) Co., Ltd. (Beijing, China).

### 2.5. Optimization of Reaction Conditions for ddPCR

The ddPCR reaction system had a total volume of 20 µL, comprising 10 µL dPCR Supermix for probes (Bio-Rad, Hercules, CA, USA), 1.5 µL of both upstream and downstream primers (10 µM), 0.5 µL probe (10 µM), 4.5 µL ddH2O, and 2 µL DNA template, 2× dPCR Supermix for probes 10 µL (Bio-Rad, USA), forward primer (10 µM) 1.5 µL, reverse primer (10 µM) 1.5 µL, probe (10 µM) 0.5 µL, ddH2O 4.5 µL, and DNA 25 ng/µL 2 µL. The reaction mix was added to the middle channel of an eight-channel disposable droplet generation card, with 70 µL of droplet generation oil placed in the third row of channels. Droplets were generated using the QX200 system’s droplet generator and transferred to a 96-well PCR plate. After sealing, the plate was subjected to PCR amplification. The PCR conditions were: pre-denaturation at 94 °C for 10 min, followed by 40 cycles of 95 °C for 15 s and 56–62 °C for 90 s, with final inactivation at 98 °C for 10 min, at a heating rate of 2 °C/s. After amplification, droplet fluorescence signals were captured using a QX200 droplet reader and analyzed with QuantaSoft 1.7.4.0917 software.

### 2.6. Specificity of ddPCR for MSTN-Edited Detection

The MSTN gene is expressed in all the tissues of the animal body. To assess ddPCR specificity, MSTN plasmid DNA and genomic DNA from six non-gene-edited species (cattle, sheep, pigs, quail, chicken, and pigeons) were used as templates. ddH2O served as a blank control, allowing evaluation of the detection method’s specificity.

### 2.7. Linearity and Limit of Detection (LOD) of ddPCR for MSTN-Edited Detection

Appropriate-concentration plasmid DNA samples were continuously diluted in a gradient, and plasmid DNA with a copy number ranging from 5 to 16,000 copies/μL was selected as the template. Concentrations were tested in four replicates. Copy number values were counted and a standard curve was drawn, plotting the theoretical copy number on the x-axis and the total copy number of the reaction system on the y-axis. This determined the linear correlation range for quantitative detection. The reproducibility of the ddPCR method was verified through calculating the relative standard deviation among repetitions.

### 2.8. Repeatability and Limit of Quantitation (LOQ) Testing

We performed ddPCR detection on six plasmid DNA samples with concentrations ranging from 10 to 16,000 copies/μL, repeating the detection four times for each sample. Based on the detection results, we calculated the relative standard deviation, analyzed the stability of the detection method, and determined the quantification limit of the detection method.

### 2.9. Practical Sample Testing

We used a blood genomic DNA extraction kit (TIANGEN: DP348-03) to extract the gene-edited cattle genomic DNA from 11 actual strain samples provided by Inner Mongolia University. ddPCR was performed to evaluate the effectiveness of the method in the actual sample detection. One of the genomic DNA samples and one of the plasmid DNA samples were diluted to similar copy numbers and then diluted into 5%, 1%, 0.1% MSTN-edited cattle samples and plasmid DNA samples for testing. We compared the interchangeability of this method in the detection of actual samples and plasmid DNA.

## 3. Results

### 3.1. Screening of Primers and Probes

By designing three sets of primer–probe combinations, F1R1-P1, F2R2-P2, and F2R3-P2 (Table 1), and using plasmid DNA, wild-type cattle DNA, sheep DNA, and ddH2O as a blank control as templates, the optimal primer–probe pair was screened using qPCR detection. The results are shown in Figure 2. In the results for primer–probe F1R1-P1, only plasmid DNA showed an amplification curve, with a CT value around 20. For the combinations of primer–probe F2R2-P2 and F2R3-P2, all three DNA templates displayed typical amplification curves with CT values all within 25, indicating that the amplification results for wild-type cattle DNA and sheep DNA were not false positives. Therefore, these two primer–probe sets did not possess specificity. The above results indicate that primer–probe F1R1-P1 can specifically recognize the target DNA, demonstrating specificity, and is the optimal primer–probe pair for this experiment.

### 3.2. Establishment of ddPCR Method

In ddPCR detection, the annealing temperature and the primer–probe concentration ratio can affect the accuracy of the results. Therefore, to obtain the optimal reaction conditions, eight annealing temperatures (55, 56, 57, 58, 59, 60, 61, and 62 °C) and four primer–probe concentration ratios were set for ddPCR detection. As demonstrated in Figure 3A, the discrepancy between the fluorescence signals of positive and negative droplets is maximized at 56 °C. Consequently, the number of positive droplets is at its zenith, signifying that the number of amplification products is also at its zenith. Moreover, the disparity in fluorescence amplitude between the positive control (blue fluorescence signal) and the negative control (gray fluorescence signal) is also at its zenith. Consequently, 56 °C was identified as the optimal annealing temperature for the ddPCR reaction procedure. The results of the annealing temperature test indicated that the difference between positive and negative droplets was the greatest at 56 °C, with good dispersion. Therefore, 56 °C was determined to be the optimal annealing temperature for the ddPCR reaction program. As demonstrated in Figure 3B, the results of the primer–probe concentration ratio test showed that in the reaction system, when the primer concentration ratio was 750:250 nM, the separation of positive and negative droplets was the best and the fluorescence signal was stronger. Therefore, the optimal primer–probe concentration ratio for the reaction system was determined to be 750:250 nM.

### 3.3. Analytical Specificity of the ddPCR Assay

Given the high degree of conservation of the MSTN gene across animal species, the objective of this analysis was to assess the specificity of the ddPCR method. To this end, plasmid and genomic DNA from six non-gene-edited animals were selected as templates, and the results are presented in Figure 4. Only the plasmid DNA yielded positive microdroplets, while the DNA of the other six animals exhibited negative results. These findings demonstrate that the established ddPCR assay exhibits good specificity.

### 3.4. Linearity and Limit of Detection Testing

The plasmid DNA was detected at concentrations ranging from 5 to 16,000 copies/μL, and the results are presented in Table 2. At a copy number of 5 copies/μL, the mean detection value was 6.75 copies/μL, with an RSD of 32.85%. Furthermore, a positive signal was detected with a high degree of stability. It was predicted that the limit of detection (LOD) of the method would be 5 copies/μL. When the template concentration was between 10 and 16,000 copies/µL, a standard curve was plotted by taking the logarithm of the theoretical copy number as the horizontal coordinate and the logarithm of the detected copy number as the vertical coordinate (see Figure 5). The resulting linear equation was y = 0.9904x + 0.24898, R^2^ = 0.9981, indicating that the template concentration was within the range of 10 to 16,000 copies/μL. The logarithm of the measured value of the ddPCR method exhibited a strong linear relationship with the logarithm of the theoretical copy number.

### 3.5. Analytical Repeatability and LOQ

Six plasmid DNA samples with serially diluted concentrations were analyzed multiple times using the ddPCR method, and the relative standard deviation (RSD) was calculated based on the detection results. The findings are summarized in Table 2. Within the concentration range of 10 to 16,000 copies/μL, the RSD ranged from 1.89% to 7.22%, indicating that the ddPCR method exhibited excellent repeatability and provided stable and reliable measurements across replicates. At a concentration of 10 copies/μL, the mean detected concentration was 13.25 copies/μL with an RSD of 7.22%, establishing the limit of quantification (LOQ) for this method at 10 copies/μL.

### 3.6. Practical Sample Testing

To ascertain the practical applicability of this method, 11 authentic samples of MSTN gene-edited cattle were selected for ddPCR. The results of the assay are presented in Figure 6, and the results of all 11 samples demonstrated positive microdroplet amplification (Figure 6A,B). Subsequently, gene-edited bovine samples at concentrations of 5%, 1%, and 0.1% as well as plasmid DNA samples were prepared for ddPCR assay. The results of the genomic DNA microtiter plots (C), copy number plots (D), and the plasmid DNA microtiter plots (E), copy number plots (F) of the two samples at three concentrations demonstrated that both samples exhibited comparable assay results and were successfully detected. The results demonstrate that both samples exhibit comparable outcomes. As the dilution of the sample gradient increased, the number of copies detected also rose in conjunction with the gradient concentration. The outcomes of the same concentration of samples with a similar number of copies detected indicate that the method is suitable for the detection of actual gene-edited bovine samples and can detect samples with a minimum concentration of 0.1%.

## 4. Discussion and Conclusions

Gene editing has revolutionized biological breeding, offering immense potential for precision breeding [23,24]. With the advancement of gene-editing tools like CRISPR and in situ editing, we can now precisely replace, insert, or delete specific bases in genomic DNA without creating double-strand breaks, significantly expanding editing capabilities. However, in situ editing still suffers from relatively low efficiency [25,26] making it challenging to quantify low-frequency mutations with traditional methods. As gene-edited products continue to develop, regulatory frameworks vary across countries. Detecting gene-edited products, especially in foods with low DNA content, remains challenging for food safety testing. Research has shown that ddPCR offers enhanced sensitivity and specificity, even when handling minute amounts of nucleic acids. The latest ddPCR technology enables absolute quantification by partitioning the reaction, making it a valuable tool for detecting genetically modified organisms (GMOs), such as genetically modified corn.

In this study, a novel ddPCR method for the detection and quantification of MSTN gene-edited cattle was developed. We established a method to detect MSTN gene-edited cattle based on the ddPCR method. This method boasts high specificity and sensitivity, with a minimum detection limit of 5 copies/μL and the ability to accurately detect samples containing 0.1% edited components. These features suggest that the method is suitable for the detection of actual gene-edited bovine samples. In comparison with qPCR, ddPCR has the capacity to produce absolute quantification without reliance on a standard curve, thereby reducing experimental error. Furthermore, ddPCR demonstrates robust tolerance to inhibitors and is well suited for the analysis of complex samples, thus positioning it for a wide range of applications in the domain of gene editing. As gene-editing technology continues to gain popularity, the demand for its product detection is anticipated to rise. This method is poised to support the future detection of gene-editing products. However, digital PCR is not without its limitations. When analyzing the resulting data, the threshold setting affects the interpretation of the results, and verification is dependent on experience or repeated experiments. The ability to detect large insertions and deletions is limited, and these must be confirmed by sequencing. ddPCR is only able to detect the editing products of known target sequences, and it cannot detect unknown editing sites or off-target effects. In order to verify the results at a genome-wide level, NGS must be used. It is important to note that the accuracy of the detection results may be influenced by gene homology and the occurrence of off-target effects in gene editing. To address these challenges, future research and applications should focus on the following: first, a thorough examination of existing methods to identify and overcome the aforementioned difficulties; and second, the cross-application of gene-editing technology with other detection technologies or disciplines, such as bioinformatics, artificial intelligence, and other relevant fields. This approach will result in the development of intelligent and automated detection systems, enhancing detection efficiency and addressing market demands while promoting the advancement of gene-editing technology.

Due to its accuracy, sensitivity, stability, and absolute quantification capabilities, ddPCR is widely used in developing nucleic acid reference materials. In the future, this method can be employed to develop standard materials for MSTN gene-edited cattle, offering accurate calibrators and positive quantitative controls for gene-edited sample detection. This will support the implementation of a quantitative identification system for gene-edited products.

## Figures and Tables

**Figure 1 biology-14-00203-f001:**
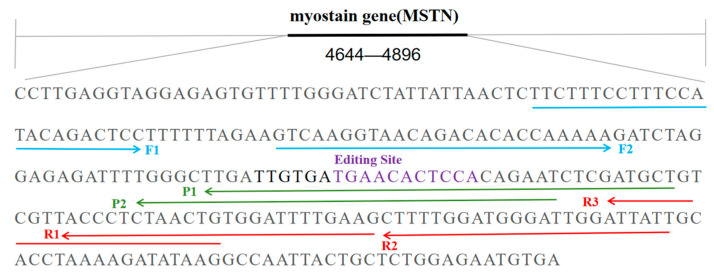
The light-blue line is the forward primer position, the red line is the reverse primer position, the green line is the probe position, and the purple font is the gene-knockout position.

**Figure 2 biology-14-00203-f002:**
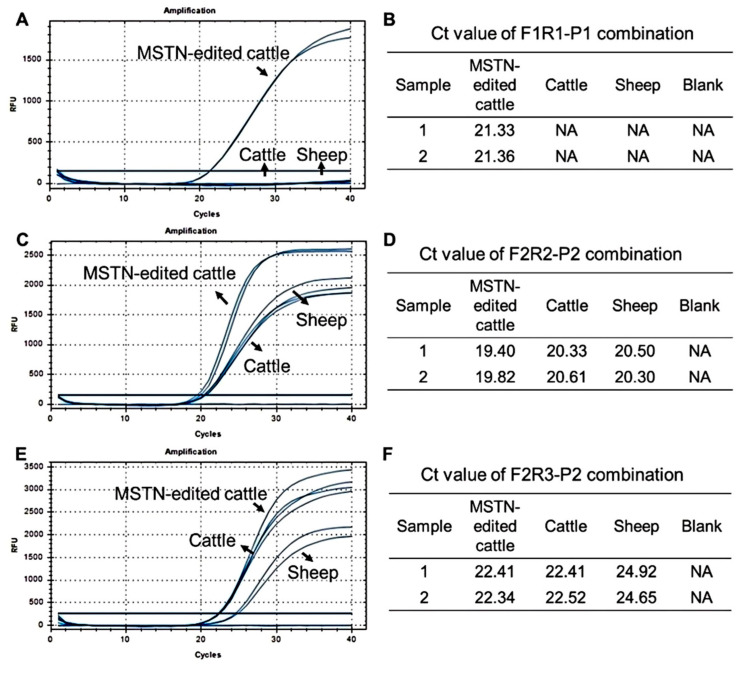
Screening of primers and probes. (**A**,**B**) F1R1-P1 combined curves and CT results; (**C**,**D**) F2R2-R2combined curves and CT results; (**E**,**F**) F2R3-P2 combined curves and CT results. NA: NA indicates that the CT value is 0.

**Figure 3 biology-14-00203-f003:**
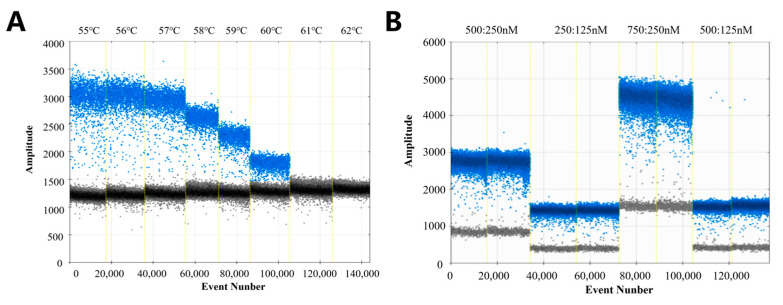
ddPCR annealing temperature and primer–probe concentration ratio optimization. (**A**) temperature set to 55, 56, 57, 58, 59, 60, 61, and 62 °C. (**B**) Tested ratios of 500:250 nm, 250:125 nm, 750:250 nm, and 500:125 nm.

**Figure 4 biology-14-00203-f004:**
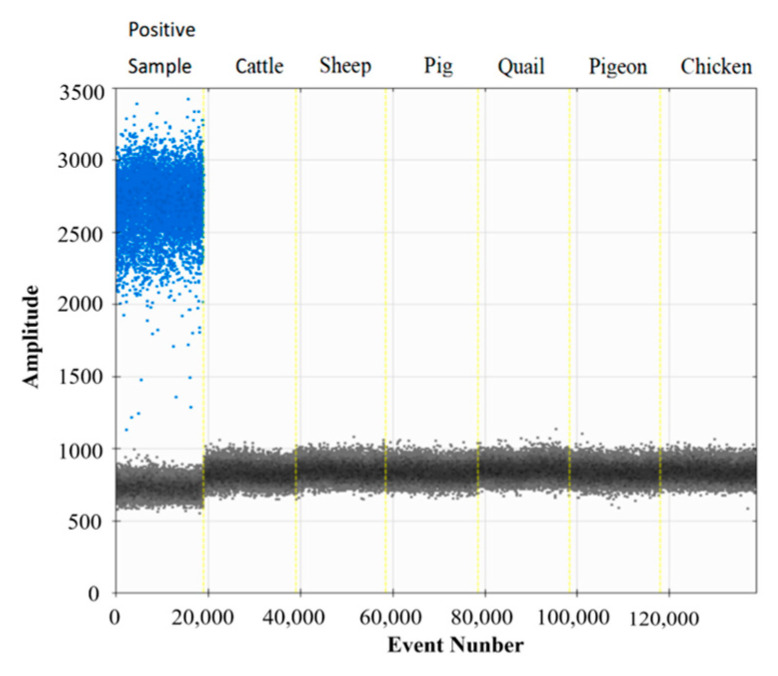
Specificity evaluation of the ddPCR method. Animal genomic DNA samples in each lane include positive sample, cattle, sheep, pig, quail, pigeon, and chicken (from left to right).

**Figure 5 biology-14-00203-f005:**
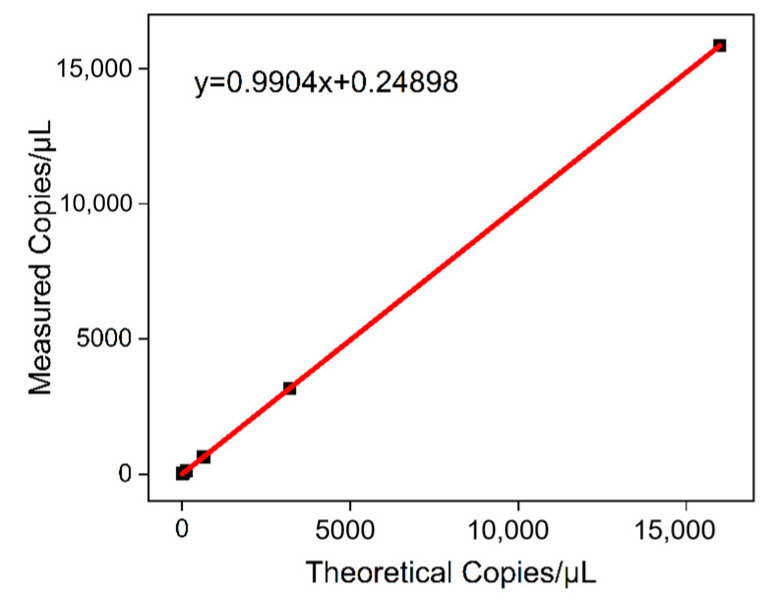
Determining the ddPCR theoretical copy number concentration (copies/µL) kinetics range by plotting the logarithmic values of expected versus measured copy number concentrations.

**Figure 6 biology-14-00203-f006:**
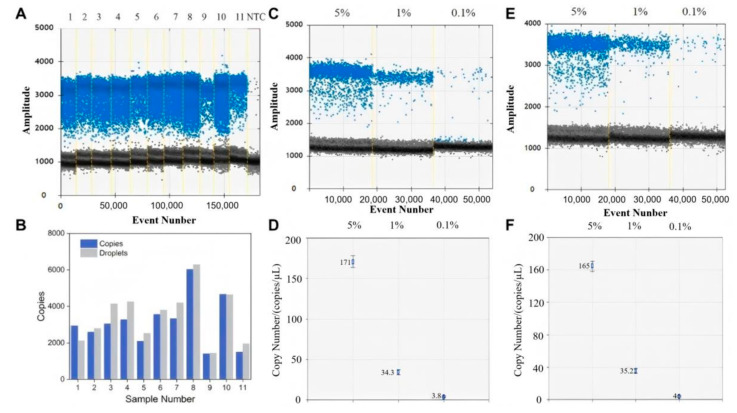
Digital PCR detects the results of the actual sample and the results of the actual and plasmid DNA samples at concentrations of 5%, 1%, and 0.1%. (**A**) Results of 11 gene-edited bovine samples; (**B**) copy number and droplet number maps of 11 gene-edited samples; (**C**) images of three actual sample droplets at different concentrations; (**D**) DNA copy number maps of three actual samples at different concentrations; (**E**) images of three plasmid DNA droplets of different concentrations; (**F**) DNA copy number maps of three plasmids at different concentrations.

**Table 1 biology-14-00203-t001:** Sequence information for primers and probes.

Name	Sequence
F1	TTCTTTCCTTTCCATACAGACTCC
R1	CTTCAAAATCCACAGTTAGAGGGT
F2	GTCAAGGTAACAGACACACCAAAAA
R2	AATAATCCAATCCCATCCAAAAG
R3	CAGTTAGAGGGTAACGACAGCATC
P1	FAM-AGCATCGAGATTCTGTCACAATCAA-BHQ1
P2	FAM-CGAGATTCTGTCACAATCAAGCCCA-BHQI

**Table 2 biology-14-00203-t002:** ddPCR reproducibility test results.

Theoretical Concentration (Copies/µL)	Mean of Detected Concentration (Copies/µL)	SD	RSD
16,000	15,850	51.23	4.8%
3200	3155	25.82	1.89%
640	634	8.66	2.15%
128	125	2.89	3.47%
25	38	0.94	5.46%
10	13.25	0.95	7.22%
5	6.75	2.22	32.85%

## Data Availability

All relevant data will be provided upon request.

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
