# Peer review of "A Novel Quantification Method for Gene-Edited Animal Detection Based on ddPCR"

_biology, 2025, doi:10.3390/biology14020203_

Round 1
Reviewer 1 Report
Comments and Suggestions for Authors
Dear َ Authors,
I hope this message finds you well. I would like to bring to your attention some points that need to be addressed in the submitted manuscript.
1. In the Materials and Methods section, instead of volume, the concentration of the reagents should be mentioned. For example, in line 113, the amount of DNA should be provided in nanograms.
2. Some of the sequences presented in Table 1 do not fully correspond with Figure 1, which displays their positions on the genome. Figure 1 needs to be corrected. Additionally, different haplotypes should be shown.
3. The DNA samples from pigs, chickens, quails, and pigeons mentioned in section 2.1—where were they used? There is no reference to these samples in the Results section.
4. The discussion is weak and should incorporate more appropriate and additional references.
Thank you for considering these suggested revisions.
Best regards
Comments on the Quality of English Language-
Reviewer 2 Report
Comments and Suggestions for Authors
The author presented a manuscript entitled "A novel quantification method for gene-edited animals detection based on ddPCR" in which a identification for gene-editing was performed. The English is good, with a few typos that need correction. The scientific methodology seems ok. The presentation and clarity of the manuscript is lacking is some aspects, mostly on the figures. The major point for revision are the figures, which are very poorly presented. They lack proper unit display, clarification and the captions and legends are horrible to read. Nevertheless, the applied methodology was an interesting read despite the poor presentation of the data in the figures.
L48 shees are sheeps mispelled I believe
L47 The phrase “the phenomenon of ‘two-muscle’” is more commonly referred to as “double-muscling.” Using a standard term would enhance clarity.
The statement “it has no effect on the survival of animals” could be oversimplified. While MSTN knockout is often described as not lethal, whether it truly has no effect on survival may need clarification or supporting evidence.
The text refers to strains provided by Inner Mongolia University.” The word “strains” is more commonly used for microorganisms or certain laboratory animal lines. For cattle, terms such as “animals,” “individuals,” or “samples” might be clearer unless there is a specific breed/line context implied.
Please clearly state the units used on the graphs and tables of Figure 2
RFU is undefined
Figure 3, 6 and 4 captions are too small and the units of the graphs are undefined
L266 Did the authors developed the method or simply applied it? Careful with this claim
Round 2
Reviewer 2 Report
Comments and Suggestions for Authors
The authors answered all the raised questions and suggestions properly